# Durable Complete Remission and Long-Term Survival in FDG-PET Staged Patients with Stage III Follicular Lymphoma, Treated with Wide-Field Radiation Therapy

**DOI:** 10.3390/cancers12040991

**Published:** 2020-04-17

**Authors:** Michael P. MacManus, Rodney J. Hicks, Mathias Bressel, Belinda A. Campbell, Andrew Wirth, Gail Ryan, H. Miles Prince, Max Wolf, Rachel Brown, John F. Seymour

**Affiliations:** 1Department of Radiation Oncology Peter MacCallum Cancer Centre, 305 Grattan St, Melbourne, VIC 3000, Australia; belinda.campbell@petermac.org (B.A.C.); andrew.wirth@petermac.org (A.W.); gail.ryan@petermac.org (G.R.); racheljeanbrown@gmail.com (R.B.); 2Sir Peter MacCallum Department of Oncology, The University of Melbourne, Parkville, VIC 3010, Australia; rod.hicks@petermac.org (R.J.H.); miles.prince@petermac.org (H.M.P.); max.wolf@petermac.org (M.W.); john.seymour@petermac.org (J.F.S.); 3Department of Molecular Imaging, Peter MacCallum Cancer Centre, 305 Grattan St, Melbourne, VIC 3000, Australia; 4Centre for Biostatistics and Clinical trials, Peter MacCallum Cancer Centre, 305 Grattan St, Melbourne, VIC 3000, Australia; mathias.bressel@petermac.org; 5Department of Haematology, Peter MacCallum Cancer Centre & Royal Melbourne Hospital, Melbourne, VIC 3000, Australia

**Keywords:** follicular lymphoma, radiation therapy, chemotherapy, rituximab

## Abstract

Advanced-stage follicular lymphoma (FL) is generally considered incurable with conventional systemic therapies, but historic series describe long-term disease-free survival in stage III disease treated with wide-field radiation therapy (WFRT), encompassing all known disease sites. We report outcomes for patients staged with ^18^F-fluorodeoxyglucose positron emission tomography (FDG-PET) and treated with CT-planned WFRT, given as either comprehensive lymphatic irradiation (CLI) or total nodal irradiation (TNI). This analysis of a prospective cohort includes PET-staged patients given curative-intent WFRT as a component of initial therapy, or as sole treatment for stage III FL. Thirty-three PET-staged patients with stage III FL received WFRT to 24–30Gy between 1999 and 2017. Fifteen patients also received planned systemic therapy (containing rituximab in 11 cases) as part of their primary treatment. At 10 years, overall survival and freedom from progression (FFP) were 100% and 75%, respectively. None of the 11 rituximab-treated patients have relapsed. Nine relapses occurred; seven patients required treatment, and all responded to salvage therapies. A single death occurred at 16 years. The principal acute toxicity was transient hematologic; one patient had residual grade two toxicity at one year. With FDG-PET staging, most patients with stage III FL experience prolonged FFP after WFRT, especially when combined with rituximab.

## 1. Introduction

Follicular lymphoma (FL) is a common mature B-cell malignancy, characterised by a long natural history, remarkable sensitivity to radiation therapy (RT) [1] and high response rates to immunochemotherapy [2]. Grade 3B follicular lymphoma is biologically akin to the aggressive diffuse large B-cell lymphoma [3] and is managed accordingly. All other histologic grades of FL are known to be potentially curable by RT [4,5,6] or RT plus chemo-immunotherapy, when limited to stages I and II [7,8]. Stage III FL, where disease is confined to nodal sites on both sides of the diaphragm, but without bone marrow or visceral involvement, is typically grouped with stage IV in the “advanced disease” category in current management paradigms and in modern clinical trials [9,10]. Advanced FL is widely considered to be incurable with any current therapy, except allogeneic hematopoietic stem cell transplantation, which is applicable in an extremely small proportion of patients [11]. Complete remission is usually achieved following induction therapy with immuno-chemotherapy [12,13,14] and progression-free survival can be improved by maintenance CD20-antibody, rituximab [15] or obinutuzumab [16]. In selected patients, durable remissions can be attained with high dose systemic therapy and hematopoietic stem cell transplantation [17,18] but it is unclear whether this approach is curative [19]. Despite initial excellent responses, most patients with advanced FL eventually experience relapse [20]. Subsequent remissions can often be attained with further systemic therapies, but these are typically of progressively shorter durations and lymphoma is the most common cause of death [21]. 

Before effective systemic therapy became available, RT was the most widely used therapeutic modality for indolent lymphomas. For patients with stage III FL, the achievement of long term disease-free survival has been consistently reported in a proportion of patients after wide-field RT (WFRT) to nodal groups above and below the diaphragm, either as a single modality [22,23,24,25,26], combined with chemotherapy [27], or following relapse after chemotherapy [28]. In a historic series from Stanford [24,26], MD Anderson Cancer Center [29], Wisconsin [23], Florida [22,30] and a more recent series from Wurzburg [25], a substantial proportion of patients with stage III FL have consistently attained long-term progression-free survival (PFS) and were apparently cured after WFRT. Barriers to the more widespread utilisation of such WFRT approaches include the inability to confidently exclude low volume or focal marrow disease, imprecision and low sensitivity for the detection of small nodal or other extra-medullary sites of disease involvement below the detection limit of CT scanning, and concerns regarding the potential late toxicities of higher dose large field RT. Ensuring adequate sampling and rigorous pathological scrutiny, including immuno-histochemistry, of multiple levels have enhanced the detection of low level marrow disease [31] and modern PET-CT has enhanced systemic disease detection [32,33,34].

With the advent of effective (albeit non-curative) systemic therapy regimens, and due to toxicity concerns and technical complexity, WFRT approaches have largely been superseded by systemic therapy or by “watchful waiting” [35] as initial management strategies for patients with stage III FL. Our multidisciplinary group has continued to offer curative–intent WFRT to patients with rigorously staged IIIA FL, who met predefined eligibility criteria. When ^18^F-fluorodeoxyglucose positron emission tomography (FDG-PET) became available at our centre, it was routinely used for staging patients with FL, along with bone marrow biopsy. PET frequently upstages apparently localized FL to advanced disease [32], and by identifying patients with disease that is too extensive for potentially curative RT, patient selection, and thereby the results of RT, can be improved. In this paper, we report disease control and toxicities for patients with stage III FL treated with curative-intent WFRT in the PET era. 

## 2. Results

Between July 1999 and December 2017, 36 patients commenced WFRT in the form of either comprehensive lymphatic irradiation (CLI, *n* = 30) or total nodal irradiation (TNI, *n* = 3), for stage III follicular lymphoma. CLI and TNI are described in the methods section. Two patients were ineligible for analysis, because the RT was delivered for management of relapsed disease after prior therapy (RT and chemotherapy respectively) and a third patient did not have a pre-treatment PET scan, leaving 33 eligible patients. The demographics of these 33 analysed patients are shown in Table 1. The median age was 50 years and 17 (51%) were female. All patients had at least one FDG-avid site of disease on PET imaging. The maximum number of Ann Arbor sites involved by lymphoma was eight and the largest tumour diameter was 8.5 cm. The follicular lymphoma international prognostic index (FLIPI) scores were 1 *n* = 18, 2 *n* = 14 and 3 *n* = 1.

Because of concerns that future intensive chemotherapy may not be deliverable due to marrow toxicity if an early relapse occurred after RT, initially all patients were routinely offered pre-emptive peripheral blood hematopoietic stem cell harvesting, with G-CSF, and often cyclophosphamide for mobilisation, and several patients also received rituximab for the putative purging of potential sub-clinical diseases at this time [36]. An ongoing evaluation revealed that stored stem cells had not been used, so this practice was ceased after 23 patients had been harvested. Table 2 illustrates the regimens used for harvesting stem cells. In seven cases, rituximab was given at the time of stem cell harvesting. In addition to TNI or CLI, 15 patients (45%) received additional systemic therapy, either as adjuvant treatment or as part of the stem cell harvesting process. Five patients had multiagent systemic therapy, either before, or both before and after, RT, as part of a planned combined modality approach. Three further patients had single agent rituximab (up to four cycles), either before or after RT. 

After a response assessment with PET and/or CT imaging, patients were generally reviewed at three-monthly intervals for two years, six-monthly until five years and annually thereafter. Four patients were lost to follow up for geographic reasons (*n* = 3), or because of psychiatric disorder (*n* = 1). Long-term survivors were monitored for relapse, development of any second malignancy and potential late toxicities.

### 2.1. Radiation Therapy Delivery and Acute Toxicity

Thirty-two (97%) patients completed the prescribed radiotherapy course. There was one patient (later diagnosed with idiopathic thrombocytopenic purpura) who ceased abdominal RT due to thrombocytopenia at 21Gy. Non-hematologic acute toxicities of RT > grade 1 were all grade 2: nausea/vomiting (*n* = 4), diarrhoea (*n* = 3), xerostomia (*n* = 7), mucositis/esophagitis (*n* = 7), skin (*n* = 3), proctitis (*n* = 1). 

The hematologic toxicities are summarized in Table 3. All patients had at least grade 1 hematologic toxicity. The major hematologic consequence of RT was thrombocytopenia, causing interruptions in RT in three cases, including the above patient, who prematurely ceased abdominal RT. Although lymphopenia occurred in all patients and moderate neutropenia was seen in the majority, only one significant infective episode occurred during treatment (dermatomal herpes zoster). A single haemorrhagic event occurred; bleeding from haemorrhoids. 

### 2.2. PET Response Assessment

Thirty patients had FDG-PET for response assessment, and all (100%) showed a complete metabolic response (CMR). These assessments pre-dated the adoption of the Deauville scale [37], and would be consistent with the Deauville responses of 1–2. In two cases, PET response was assessed after neoadjuvant systemic therapy, before RT. An example of PET response assessment is shown in Figure 1. 

Maximum intensity projection (MIP) FDG-PET images of a patient with stage III follicular lymphoma, before (left panel) and after (right panel) WFRT as a single modality. The left panel shows FDG avid disease at multiple sites, including cervical, axillary, mediastinal, iliac and inguinal lymph nodes. The right panel shows a complete metabolic response, with reduced marrow activity reflecting radiation treatment volumes.

### 2.3. Overall Survival and Freedom from Progression

Individual patient outcomes are shown in the event history chart (Figure 2). The study close-out date was April 11th, 2018. Median follow up was nine (range 0.5–18.5) years. OS was 100% at 10 years and only one patient has died (in long-term remission after PBSCT aged 82). Freedom from progression (FFP) (Figure 3) for the entire cohort was 79% at 5 years (95% confidence interval 65–96) and 75% at 10 years (95% CI 60–93%). The factors potentially associated with FFP are shown in Table 4. The delivery of any systemic therapy (*p* = 0.002, Figure 4) or any rituximab (*p* = 0.025, Figure 5) was associated with superior FFP. No rituximab-treated patient had yet had a progressive disease and only one patient treated with systemic therapy without rituximab had progressed. Patients with a FLIPI score of 1 had fewer progressions than those with a score > 1 (HR 3.41, CI 0.84–13.76, *p* = 0.086).

### 2.4. Management after Progression

By the last follow-up, nine patients had manifest disease progression. In two cases, multifocal low volume relapse was managed expectantly. In two cases, with isolated relapses in the orbit and peri-renal areas respectively, CMR was attained with salvage local RT. The latter patient later relapsed with DLBCL, and after R-CHOP induction, underwent peripheral blood stem cell transplantation (PBSCT) and was in remission at last follow up > 4 years later. Two further patents had transformation to DLBCL at first relapse and were treated with salvage R-CHOP and PBSCT. Both attained sustained complete responses and one of these died without relapse. The remaining three patients all had multifocal relapses and received salvage immunochemotherapy (FCR *n* = 2, R-CHOP *n* = 1). In all cases, salvage chemoimmunotherapy was completed on schedule, with no more than the expected haematological toxicity, and in all cases CMR was attained.

### 2.5. Subacute and Late Toxicities of Radiation Therapy

Two patients had significant radiation pneumonitis of grade 2 and 4 respectively, after completion of RT. Both recovered fully, although one patient required hospital admission, and both required steroids. One patient developed renal artery stenosis at five years, within the RT field, but of uncertain relation to therapy. One patient had treatment-related bilateral avascular necrosis of the hips, requiring total hip replacements. One patient developed severe fatigue and depression after therapy and another had an anxiety disorder. One patient developed shingles after RT and two developed hypothyroidism, requiring replacement therapy. None of the patients developed any clinically significant renal parenchymal dysfunctions.

### 2.6. Second Malignancies

Five new neoplasms were diagnosed after the commencement of RT. One case each of prostate cancer (metastatic to bone), mature teratoma of the testis, melanoma of the forearm and squamous carcinoma of the scalp occurred, all outside the RT fields. One case of follicular thyroid cancer occurred within the RT field, and was successfully treated with resection. There were no cases of leukemia or myelodysplasia.

## 3. Discussion

PET-staged patients with stage III FL, treated with CT-planned WFRT, experienced exceptional long-term overall survival in this report. The majority of patients were free from relapse at 10 years, and in the small subset of patients who received any rituximab in addition to WFRT, no relapses had been observed at last follow up. No lymphoma-related deaths were recorded and only one death occurred from unrelated causes. These results not only confirm the existing, but infrequently cited, literature showing long-term disease control with WFRT in stage III FL, but they surpass the outcomes reported in those historic reports. As in earlier series, very prolonged survival without relapse could be considered consistent with “cure” in cases, where patients in long-term remission survive to normal life expectancy. 

The favourable long-term outcomes reported in this series are attributable to several factors, including careful patient selection, PET scanning excluding patients with occult systemic disease or who were too advanced for treatment with RT, and the incorporation of PET and CT information into RT planning, ensuring that any geographic miss of gross disease was avoided. PET also directed the biopsy of lesions with high standardized-uptake values to exclude aggressive lymphoma. The frequent addition of systemic therapy in this series, especially rituximab in combination with RT, was associated with a remarkably low relapse rate, consistent with “spatial cooperation”, where gross disease is controlled by RT and occult disease controlled by systemic therapy. These results are consistent with the randomized phase III TROG 99.03 [7] and the prospective phase II MIR trial [38] in stage I-II FL, in which the combination of RT to all sites of gross disease with rituximab-containing systemic therapy was associated with very high rates of durable complete remission. 

A 100% rate of CMR on post-treatment PET scanning was expected, given the known very high radiosensitivity of FL. Although RT has long been considered an exclusively local therapy, there is emerging evidence that in FL, local RT may be associated with the regression of distant disease in the absence of other treatment (MacManus MP et al., in press). The so-called “abscopal” effect of RT may be more common in FL than previously known and is considered to be an immunologically mediated phenomenon. The effectiveness of the combination of local RT and anti-CD20 antibody therapy in the TROG 99.03 and MIR trials in early stage FL may be derived from an interaction between RT, immunotherapy and the host immune system. Such an interaction is also plausible, but unproven, in the stage III setting. Because isolated local treatment failure is unusual with RT alone, it is unlikely that the efficacy of RT combined with rituximab is due to a local radiosensitizing effect. 

WFRT has largely been abandoned in stage III FL, because of its technical complexity, concerns about toxicity and the increasing effectiveness of systemic therapies, especially immuno-chemotherapies including anti-CD20 antibodies. Another concern is that salvage systemic therapy might not be deliverable after WFRT due to myelotoxicity, especially for relapses in the first year after treatment. Reassuringly, in our series, the relapse rate was very low and those patients who required systemic therapy for relapse, including those treated aggressively for transformation to aggressive lymphoma, were able to receive full dose chemotherapy without delays due to myelotoxicity, and even subsequent successful stem cell harvesting. 

The results reported here suggest that WFRT should at least be considered as a viable treatment option for selected patients with stage III, especially those who wish to pursue a “curative” intent therapeutic approach and those who refuse chemotherapy. The ideal candidate patients would have an extent of disease that would be readily treatable within the standard TNI or CLI volumes and would have a life expectancy exceeding 10–15 years in the absence of lymphoma. The long natural history of FL means that a prolonged “lead time” would be required before any potential survival benefit of long-term disease control could be realised by patients, given the excellent short and medium term PFS results that are attainable with modern systemic therapies. Patients treated with WFRT must undergo several months of daily treatment for a potential future benefit, when they could, in some cases, undergo initial management with “watchful waiting”, or proceed directly to immuno-chemotherapy. To effectively compare RT or RT plus anti-CD20 antibody therapy with current management approaches would require a prohibitively large randomized controlled trial with prolonged (> 10 years) follow up. Short-term results would be expected to be similar, even if there was a late benefit from RT.

In this series, hematologic toxicity was almost universal during the final phase of treatment and was prolonged in a few cases, especially thrombocytopenia. Nevertheless, by one year, blood counts were normal in virtually all patients. There were no significant bleeding events or bacterial infections during RT and only one significant viral infection (herpes zoster). Salvage therapies were delivered without unusual myelotoxicity in those few patients who relapsed. Prophylactic storage of hematopoietic stem cells proved unnecessary, because no autologous stem cell transplants were performed using these products, as the few transformation events seen were quite late (all >4 years after RT) and stem cells were successfully harvested in those few patients at that time. Acute toxicities other than myelosuppression included fatigue and diarrhoea, responsive to loperamide or similar agents. Nausea was largely prevented by the prophylactic use of ondansetron. Although short-term RT-related toxicities may be greater than would be expected with some first-line systemic therapies, systemic therapies are not without toxicity, and with each relapse, cumulative toxicities increase. Over the lifetime of the patient, the toxicity of wide WFRT may actually be significantly less for many patients than the cumulative toxicities of sequential systemic therapies, if RT is able to provide durable disease control. However, in the absence of randomized trials, it cannot be known that the favourable outcomes that we report here could not have been achieved with systemic therapy in a similarly selected cohort of stage III FL patients. At present, the literature contains little information on outcomes for stage III FL treated with modern systemic therapy regimens, because these cases are reported combined with stage IV patients in the “advanced” disease category.

The therapeutic approach reported here may not represent the optimum way to utilise RT in stage III FL. There has been a long-established trend towards eliminating the use of RT in a wide range of clinical scenarios in lymphoma, to prevent late toxicities. In those diseases where RT is still used, such as in early stage FL, RT is now given in lower doses and to smaller volumes than in the past [5]. These approaches have minimized the potential for early and late effects of RT. The TROG 99.03 trial [7] showed that in stage I-II FL, local RT could reliably eradicate local disease, and that systemic therapy could control occult distant disease in a majority of cases, thereby achieving very favourable long-term freedom-from relapse results. The combination of chemoimmunotherapy and local RT may also prove to be effective in preventing relapse in stage III FL. Information from the FoRT [39] trial suggests that 24 Gy is a high enough dose in FL and that very low dose RT (4Gy in 2 fractions) can control local disease without toxicity in a high proportion of cases. An alternative approach to combining RT and chemoimmunotherapy in FL stage III might involve the addition of very low dose involved-site RT to all lymphoma lesions visualized on PET. Despite the increasing efficacy of modern systemic therapies in advanced FL, not all patients achieve complete remissions, and many patients who do experience remission will relapse at initially involved sites of disease. A modern combined modality approach in stage III, with optimum systemic therapy and smaller volume, low dose conformal RT may have the potential to improve outcomes in FL with little, if any, increase in toxicity. 

## 4. Materials and Methods

This study was carried out following the rules of the Declaration of Helsinki of 1975 (https://www.wma.net/what-we-do/medical-ethics/declaration-of-helsinki/), revised in 2013. Patient data for this Peter MacCallum Cancer Centre ethics-committee approved study (approval 18_87R) were extracted from a prospective database established in 1999. Because this study reports an analysis of de-identified data from a clinical database, individual patient consent was not required according to institutional policy. Treatment details, therapeutic response information and follow-up data were available. The completeness of the patient cohort was validated by cross-referencing the hospital radiotherapy treatment database with a state-wide cancer registry, assuring complete ascertainment. Laboratory data, including information on toxicities, and blood counts were extracted retrospectively from hospital records. All potential treatment candidates were discussed at weekly multidisciplinary meetings attended by hematologists, radiation oncologists, imaging specialists and members of other relevant disciplines. Patients with an anticipated life expectancy of >10–15 years independent of their FL were offered WFRT as one of a range of available management options, that could include watchful waiting or immediate systemic therapy.

Patients were informed that RT was the only therapeutic option known at the time to be associated with a significant potential for cure. Although RT alone was the preferred option of the multidisciplinary group overall, some hematologists in our group preferred to combine WFRT with systemic therapy. All PET-stage III patients treated with wide field RT or RT plus systemic therapy are included in this report. 

### 4.1. Inclusion Criteria

Follicular lymphoma grade I–IIIa; bone marrow aspirate and trephine without morphological evidence of involvement by lymphoma, after comprehensive examination of adequate sampling (at least 3 levels of a 20 mm or greater length trephine core); stage III disease after FDG-PET staging; prescribed treatment with curative intent WFRT in the form of CLI or TNI as defined below; treatment commenced between July 1999 and December 2017.

### 4.2. Exclusion Criteria

WFRT given as salvage for relapsed disease; FDG-PET staging not performed; bulk > 10 cm; disease distribution, including sites more peripheral than axillae or groins (e.g., epitrochlear, popliteal).

### 4.3. PET Imaging

Before 2002, patients underwent FDG-PET scans on a GE Quest 300-H scanner, (UGM Medical Systems Inc., Philadelphia, PA, USA) and separately acquired CT scans. From 2002, patients underwent imaging on a combined Discovery LS PET/CT scanner (GE Medical Systems Milwaukee, WI) and subsequently on Siemens Biograph 64, GE Discovery 690 or GE Discovery 710 PET/CT systems. Standardized whole body image acquisition protocols were used and scans were read qualitatively. Patients with accessible highly FDG-avid lesions (SUVmax > 15) underwent targeted biopsies to exclude transformation to aggressive lymphoma. FDG-PET information was routinely used for radiotherapy planning, ensuring that all FDG-avid sites received at least 24–30Gy. PET imaging was also routinely performed after the completion of RT, to assess response using visual criteria [40,41].

### 4.4. Radiotherapy

Radiotherapy was delivered sequentially to supra-diaphragmatic volumes, with a minimum gap of four weeks to permit hematological recovery. The most symptomatic or largest region was treated first. In all cases, the supradiaphragmantic RT approach was similar. The infradiaphragmatic RT approach depended on the presence (CLI) or absence (TNI) of mesenteric disease, as described below. Treatment was CT-planned, with neck immobilization in a cast, and delivered in daily fractions on a linear accelerator using 6 or 18 MV photon beams. In the most recent case, intensity modulated radiation therapy (IMRT) was used. All non-bulky sites of gross disease were prescribed 24–30Gy and sites > 5cm in maximum transverse diameter were boosted to 36Gy. Wide-field RT was given in 1.5Gy fractions.

### 4.5. Supradiapragmatic RT

“Mantle” treatment volumes [42] matched the upper borders of the infradiaphragmatic volumes at the level of the top of the diaphragm. The lateral borders of each AP/PA mantle field covered the axillae. Mediastinal, hilar, infraclavicular, axillary and low-neck nodes were all included. The upper border of the mantle field was matched to the lateral opposed “Waldeyer’s” fields that covered the upper neck nodes, tonsils and nasopharynx. The mantle field was treated to 24–30 Gy, but the Waldeyer’s field was treated to 20–24 Gy if no disease was apparent.

### 4.6. Infradiaphragmatic RT

Abdominal and pelvic RT was delivered, entirely or partially, via anterior and posterior opposed fields at extended focus to skin distance (FSD). If mesenteric nodes were involved, whole abdominal and pelvic RT was delivered (CLI), often including a component from lateral fields, shaped to avoid kidneys. If mesenteric nodes were negative, an inverted-Y [42] and spleen volume were treated (TNI, example shown in Figure 6). The infradiaphragmatic field extended from the top of the diaphragm and inferiorly included the inguinal and femoral nodes, if involved, with shielding of the genitalia and the base of the bladder. After an initial phase of treatment given to 24 Gy, areas of gross disease were boosted to 30 Gy, if this could safely be accomplished. Posterior partial thickness kidney shielding was used, restricting mean kidney doses to <15 Gy. The right lobe of the liver was shielded from anterior and posterior fields after 18Gy, except in regions where gross disease would have been spared. Daily anti-emetic therapy with oral ondansetron 8mg b.i.d. was given and full blood counts were performed at least weekly during treatment. 

### 4.7. Statistical Considerations

The Kaplan–Meier method was used to estimate the time-to-event outcomes. Overall survival (OS) and freedom from progression (FFP) were measured from the first day of RT. Estimates at specific time points with associated 95% confidence intervals were reported. The exact logrank test was used to assess the prognostic value of dichotomous variables on FFP. Hazard ratios and confidence intervals were obtained from the Cox model. This was an “intention to treat” analysis and included all patients who commenced treatment. Progression was defined as the first date on which the growth of new disease was confirmed outside the radiation field by imaging or biopsy or there was regrowth or increase in size of a previously known lesion within the radiation field. Patients with fluctuating small volume lymphadenopathy of uncertain origin were considered to have disease progression only if there was a progressive increase in nodal size, or if a biopsy confirmed the presence of lymphoma. 

## 5. Conclusions

The results presented here confirm the efficacy of WFRT in stage III FL and suggest that it is a potentially curative option in appropriately selected patients. The use of PET staging, rigorous marrow evaluation and 3D conformal RT planning can produce results superior to those reported in historic series. We suggest that the addition of targeted RT to systemic therapy in stage III FL should be explored in future clinical trials. In the meantime, WFRT, with or without systemic therapy, could be reasonably offered to patients with a long life expectancy and a desire for curative-intent treatment. 

## Figures and Tables

**Figure 1 cancers-12-00991-f001:**
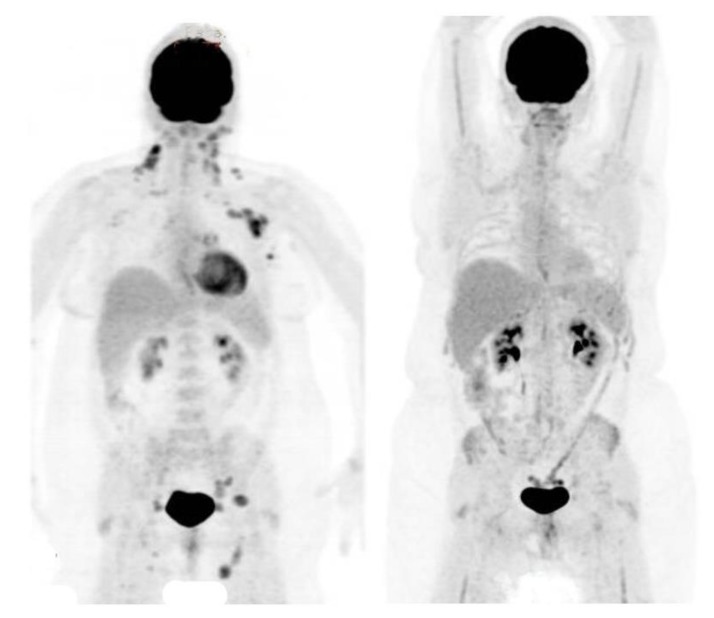
PET scans before and after wide-field radiation therapy.

**Figure 2 cancers-12-00991-f002:**
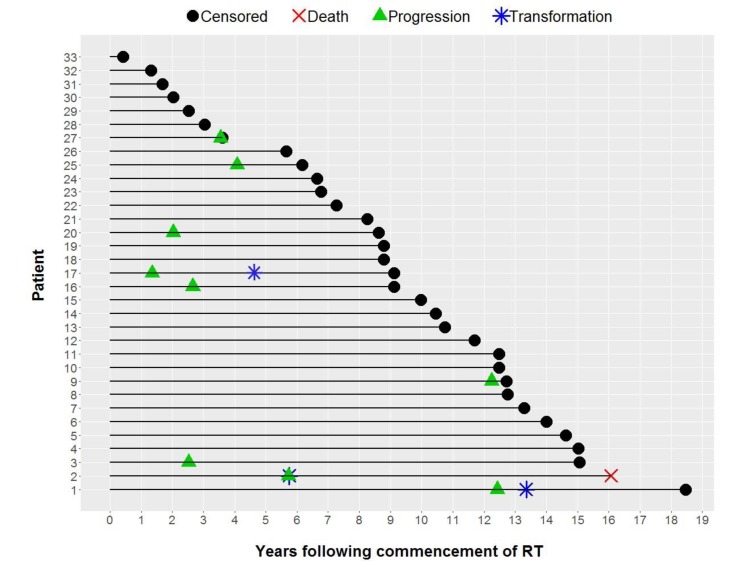
Event History Chart.

**Figure 3 cancers-12-00991-f003:**
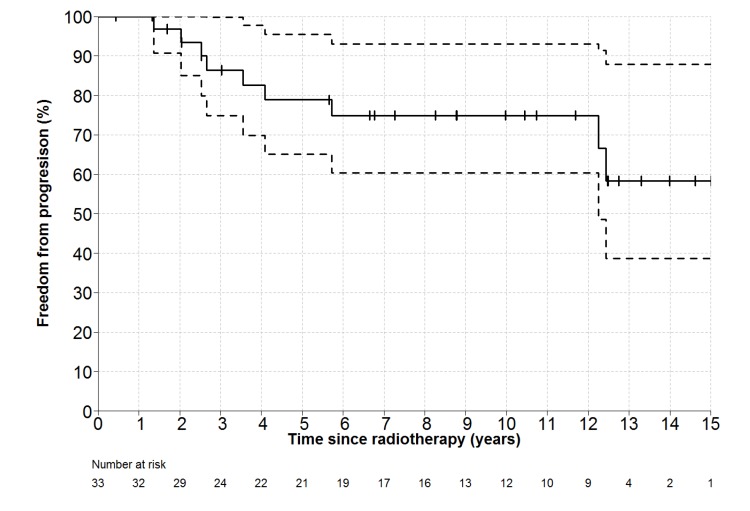
Freedom-From Progression (and 95% CI for point estimates) for all patients. Number at risk represents number of patients still under observation, without an event, at the start of the relevant time interval.

**Figure 4 cancers-12-00991-f004:**
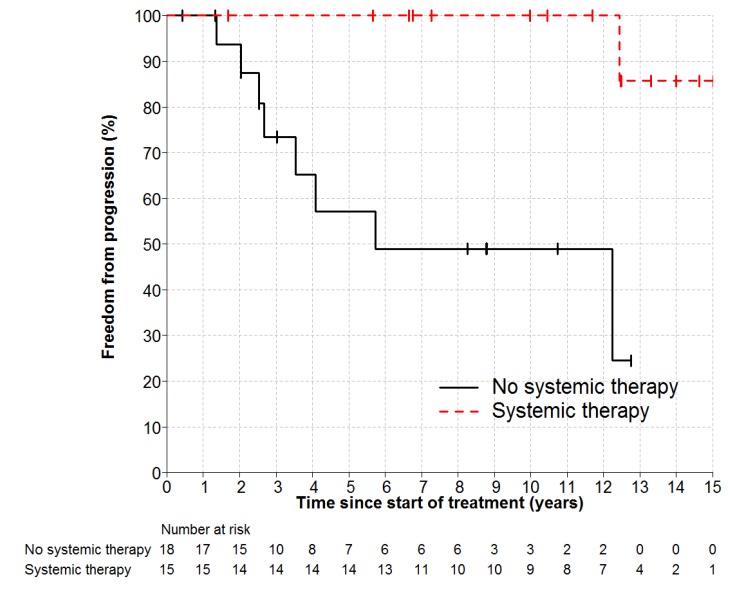
Freedom from Progression by any Systemic Therapy HR 0.1, *p* = 0.002.

**Figure 5 cancers-12-00991-f005:**
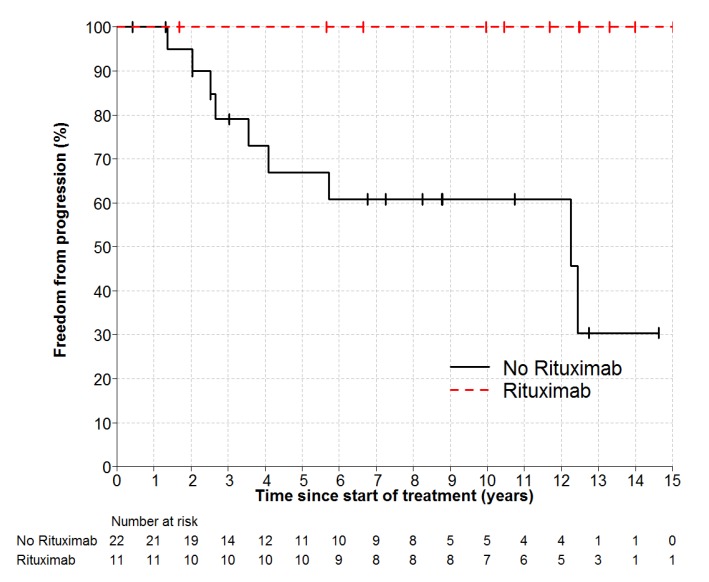
Freedom from Progression by Rituximab *p* = 0.025. HR not estimable, no rituximab patient relapsed.

**Figure 6 cancers-12-00991-f006:**
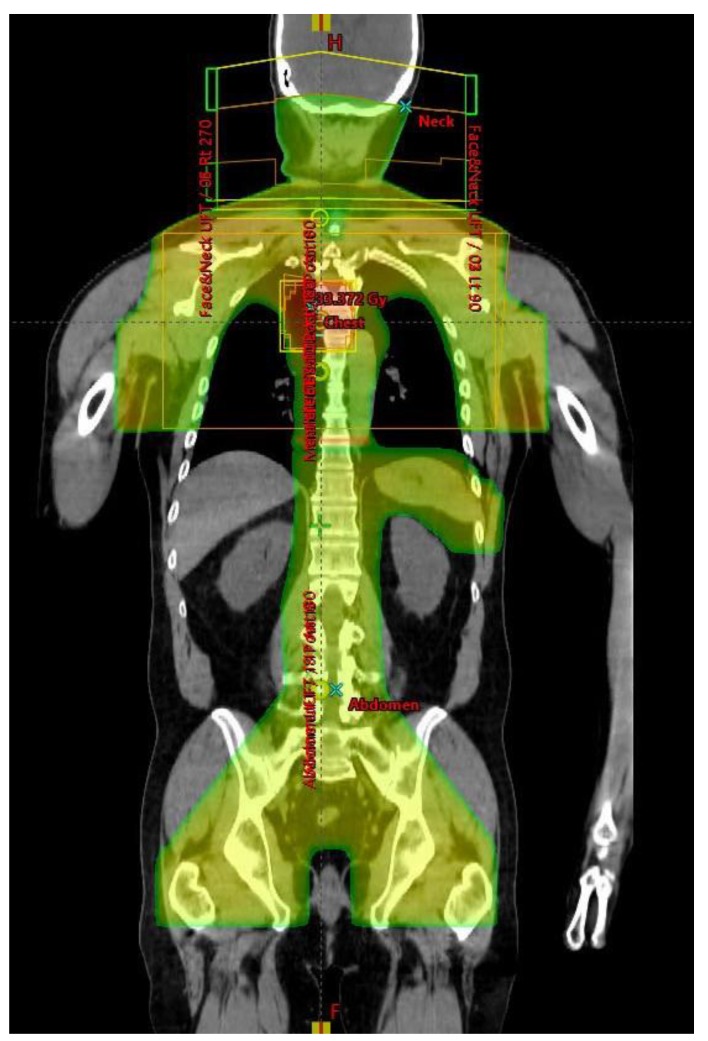
Total Nodal Irradiation dose distribution (coronal view).

**Table 1 cancers-12-00991-t001:** Demographic data.

Variable	Result
Age	
Mean (SD)	50 (9)
Median (range)	49 (2–71)
Interquartile range	44–56
Histological grade	
1–2	32 (97%)

3a	1 (3%)
Number of involved Ann Arbor Sites	
2	7 (21%)
3	6 (18%)
4	9 (27%)
5	7 (21%)
≥6	4 (12%)
Prescribed RT dose	
24Gy	4 (12%)
30Gy	29 (88%)
Maximum nodal diameter (cm)	
Mean (SD)	3 (2)
Median (range)	2.5 (1–8.5)
Interquartile range	2–3
Maximum diameter	
≤5 cm	28 (85%)
>5 cm	5 (15%)
Treatment	
RT alone	18 (55%)
RT + systemic therapy	15 (45%)
Rituximab	
No Rituximab	22 (67%)
Rituximab	11 (33%)

**Table 2 cancers-12-00991-t002:** Systemic therapies with potential anti-lymphoma effects given as a component of primary treatment or at time of stem cell harvesting. Patients receiving any anti-lymphoma systemic therapy: *n* = 15 (all shaded cells). Patients receiving any rituximab: *n* = 11 (yellow shaded cells only).

Patient ID Number	Stem Cell Harvest Mobilization	Stem Cell Harvest Purge	Systemic Therapy Pre-RT	Systemic Therapy Post RT
**Patient**	Cyclophosphamide 1.5 g/m^2^	Rituximab		
1	Yes	No		
2	Yes	No		
3	Yes	No	R-CHOP × 6	
4	No	No	CVP × 6	
5	Yes	Yes	Rituximab × 4	
6	Yes	Yes		
7	Yes	Yes		
8	Yes	No		
9	Yes	Yes		
10	No	No	R-CVP × 2	R-CVP × 3
11	No	Yes	Rituximab	
12	No	Yes		
13	No	Yes		
14	No	No	R-CHOP × 3	
15	No	Yes		Rituximab × 4

Abbreviations: R-CVP = rituximab, cyclophosphamide, vincristine, prednisolone, R-CHOP = rituximab, cyclophosphamide, doxorubicin, vincristine, prednisolone, ID = identification.

**Table 3 cancers-12-00991-t003:** Hematologic toxicity of radiation therapy. G-CSF = granulocyte colony-stimulating factor.

Toxicity	Hemoglobin	Platelets	Neutrophils
Nadir blood count Median Range	10.6 g/dL (7.4–13.8)	46 (11–238)	0.87 (0.3–3.17)
Patients with any toxicity ≥3	2/33 (6%)	16/33 (48%)	17/33 (52%)
Duration Gd ≥3 toxicity Median Range	4 days (1–8)	13 days (7–115)	12 days (1–44)
Hematological Support given	5/33 (15%) Red cell transfusion	1/33 (3%) Platelet transfusion	5/33 (15%) (G-CSF injection)
Residual Toxicity after 1y	3/33 Grade 1	3/33 grade1 1/33 grade 2 *	0

* This patient had idiopathic Thrombocytopenic Purpura.

**Table 4 cancers-12-00991-t004:** Univariable analysis for Freedom-From Progression (FFP).

Variable	Level	Number of Cases	5 years FFP (95% CI)	HR (95% CI)	*p*-Value
Any systemic therapy	No	18	57% (36–91)	1	0.002
	Yes	15	100%	0.1 (0.01–0.5)	
Any Rituximab	No	22	67% (48–93)	1	0.025
	Yes	11	100%	Not estimable	
Maximum tumour diameter	≤5 cm	28	82% (68–100)	1	0.057
	>5 cm	5	60% (29–100)	4.8 (1.1–21.6)	

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
