# Peer review of "Durable Complete Remission and Long-Term Survival in FDG-PET Staged Patients with Stage III Follicular Lymphoma, Treated with Wide-Field Radiation Therapy"

_cancers, 2020, doi:10.3390/cancers12040991_

Round 1
Reviewer 1 Report
Dear authors,
i have read your manuscript with great interest and i would like to ask you to add information and comment on if the Follicular Lymphoma International Prognostic Index was correlated in any way with your results given the large amount of clinical data you have provided and other than that i am happy with the manuscript as it is.
All the best, stay safe!
Author Response
We appreciate the kind words of the reviewer. The suggestion about the FLIPI score is an excellent one. We now give the numbers of patinets with each score (Lines 93-94) and provide an analysis of the correlation of FLIPI score with progression (lines 163-164). Patients with a FLIPI score of >1 had more progressions (HR 3.41, p=0.086).
Reviewer 2 Report
This is a well written paper dealing with the role of WFRT in patients with follicular lymphoma who had radiation therapy as the main mode of therapy.
The majority of patients actually were given additional therapy including rituximab and fifteen patients had at least 1 cycle of cyclophosphamide as preparation regimen for stem cells collection.
Please provide data regarding the differences in the outcome as presented in Fig 2 for two separate group those who had chemotherapy including intermediate dose cyclophosphamide and those who had no chemotherapy.
Page 5 line 144 please provide also the data for the 2 patients after neo-adjuvant therapy.
The Merit of the treating group is the WFRT technique .Please provide an illustration of the radiation fields and the actual technique used
Author Response
We appreciate the comments of the reviewer. We have answered each point below.
"The majority of patients were given additional therapy" . Actually the existing text indicates that 18 patients had Rt alone and only 15 had any additional therapy.
"Please provide data..." Data for the differences in outcome by systemic therapy are given in figures 4 and 5.
Data for the 2 patients given neaoadjuvant therapy are included in the anaysis of the effects of systemic therapy and their responses to neoadjuvant therapy is mentioned in the PET response section.
Illustration of RT fields. This is an excellent suggestion and we have included a new figure showing the dose distribution for a patient treated with TNI (Figure 6).
Reviewer 3 Report
Overall a well written and clearly outlined manuscript. The authors describe a clinical study involving curative treatment of aggressive stage III follicular lymphoma wide-field radiation therapy.
A] Minor comments
a) Table 4: 5-year FFP (units for this column must be mentioned)
b)It is not entirely clear what the authors mean by Number at risk under each Kaplan Meier survival plot. This needs to be clarified.
B] Major comments
1)The authors have mentioned that concurrent chemotherapy (rituximab) with IR has been very effective but the possibility of the possible role of the antibody in acting as a radiosensitizer has not been mentioned. This needs to be discussed or at least speculated on based on previous publications as most examples of radiosensitizers are small molecule drugs like cisplatin. Why does the CD20 pathway have such a profound effect on the therapeutic response seen when coupled with IR.
2)What proportion of patients in this study developed bone marrow issues as a consequence of radiation?
3)Would lower Gy but multiple cycles of treatment prove advantageous, given that most antibodies have a long circulation and tissue retention time of days as opposed to hours for small molecules.
Author Response
We appreciate the kind words of the reviewer. Each point is answered below:
Table 5 FPP, Units are now given
Numbers of patients at risk are now explained in the legend of figure 3.
Radiosensitisation by rituximab. This is an intriguing suggestion but we do not think that this is the primary effect of rituximab in this setting, given that the isolated local relapse trate for patients treated with IFRT is less than 5%. We suspect that the effect of rituximab is an immunological one, operating on microscopic residual disease. We have clarified this issue in the discussion (lines 262-263).
Proportion with bone marrow damage? All patients had some degree of haematological toxicity (even if only grade 1). This has been clarified on line 133-134.
The final suggestion concerning use of infrequent low doses of RT in the setting of a monoclonal antibody with a long half life is very interesting, but would call for speculation beyond the immediate scope of the article so we have regretfully decided not to pursue that suggestion in the text.